# The Effect of Stimulants on Nectar Composition, Flowering, and Seed Yield of Common Buckwheat (*Fagopyrum esculentum* Moench)

**DOI:** 10.3390/ijms241612852

**Published:** 2023-08-16

**Authors:** Agnieszka Płażek, Michał Dziurka, Aneta Słomka, Przemysław Kopeć

**Affiliations:** 1Department of Plant Breeding, Physiology and Seed Science, University of Agriculture, Podłużna 3, 30-239 Kraków, Poland; 2Franciszek Górski Institute of Plant Physiology, Polish Academy of Sciences, Niezapominajek 21, 30-239 Kraków, Poland; michal.dziurka@gmail.com (M.D.); p.kopec@ifr-pan.edu.pl (P.K.); 3Department of Plant Cytology and Embryology, Institute of Botany, Faculty of Biology, Jagiellonian University in Kraków, Gronostajowa 9, 30-387 Kraków, Poland; aneta.slomka@uj.edu.pl

**Keywords:** amino acids, common buckwheat, nectar, plant stimulants, polyamines, seed setting

## Abstract

Common buckwheat is a valuable plant producing seeds containing a number of health-promoting compounds and elements. Buckwheat does not contain gluten and is characterized by an excellent composition of amino acids. This species is also a melliferous plant. Despite many advantages, the area of buckwheat cultivation is decreasing due to unstable yields. One of the reasons for low seed yield is its sensitivity to drought, high temperatures, and assimilate deficiencies. These factors have a significant impact on the nectar composition, which is important for visiting pollinators and thus for pollination. High temperature during flowering increases the degeneration of embryo sacs and embryos, which is high anyway (genetic determination) in common buckwheat. This phenomenon seems to be unbreakable by breeding methods. The authors aimed to determine whether stimulants commonly used in agriculture could increase the seed yield of this plant species. The aim of the work was to choose from eight different stimulants the most effective one that would improve the seed yield of two accessions of common buckwheat by increasing the efficiency of nectar production and reducing the number of empty seeds. The plants were sprayed at either the beginning of flowering or at full bloom. The content of sugars and amino acids was higher in the nectar produced at the beginning of flowering. The nectar of both lines included also polyamines. The level of sugars in the nectar increased mainly after spraying with the stimulants in the second phase of flowering. A positive correlation between the total amount of sugars and amino acids in the nectar and seed yield was found. All the stimulants used reduced the number of empty seeds in both accessions. Seed production in the PA15 line increased significantly under the influence of all stimulants used at the beginning of flowering, and the most effective were ASAHI SL and TYTANIT^®^.

## 1. Introduction

Food production is increasingly constrained by climate change causing the Earth’s temperature to rise. Prolonged heat combined with drought causes huge losses in agricultural crops [1]. Modern agriculture must therefore focus on the production of new crop cultivars that are more resistant to unfavorable environmental conditions or are supported by appropriate cultivation techniques. 

The second problem faced by the modern world is civilization diseases, including food allergies to many substances, for example, proteins including gluten, causing skin diseases, asthma, and intestinal disorders [2]. These factors encourage scientists to search for new cultivars of crops not only more resistant to environmental stresses but also serving as alternative food sources free of allergenic factors and rich in vitamins, minerals, and antioxidants. These crops include, e.g., common buckwheat (*Fagopyrum esculentum*) and Tartary buckwheat (*F. tataricum*) [2].

Common buckwheat from the *Polygonaceae* family belongs to pseudo-cereals. It is cultivated mainly due to its favorable chemical composition of seeds, especially a very high content of starch, minerals, vitamins, rutin, and dietary fiber, unique amino acid composition, especially an abundance of lysine and arginine, antioxidants, and a lack of gluten [2,3,4]. Buckwheat seeds contain high amounts of macro-elements (phosphorus, potassium, calcium, and magnesium) and very valuable microelements, especially iron and zinc [5]. Moreover, common buckwheat has some medicinal properties due to its high content of phenolic compounds including rutin [6,7]. Buckwheat nectar is a source of valued honey with anti-inflammatory and antioxidant properties [8,9].

The largest buckwheat producers in 2022 were Russia, China, France, Poland, Ukraine, and USA [10]. In 2021, the area of buckwheat cultivation in the world was 1,988,545 ha with a seed yield of over 1,875,067 tones [11]. The area of buckwheat cultivation varies every year due to unstable seed yield. Common buckwheat is sensitive to unfavorable environmental factors, especially to frost, low or high temperature, and drought [3,12,13]. Its seed yield is also unstable due to specific flowering biology, i.e., heterostyly. Common buckwheat produces two flower morphs: Pin—flowers with long pistils and short stamens, and Thrum—flowers with short pistils and long stamens. Fertilization requires cross-pollination between these different types of flowers, and consequently, common buckwheat demonstrates strong self-incompatibility [14,15,16]. An additional problem is that a flower is available for fertilization for only one day [14,17,18]. Common buckwheat shows also high female sterility, even though its pollen demonstrates high viability amounting to over 90% [19]. The high percentage of ovule and embryo degeneration has a genetic background [8,12,14,15] and is increased by assimilate deficiencies when the plant ages [20]. High temperature increases disturbances in embryo sac development [21,22], which may additionally be affected by increasingly hot summer months. Embryo death was also observed as a result of starvation due to the insufficient distribution of assimilates during prolonged flowering and simultaneous ongoing filling the seeds [23]. Buckwheat has an indeterminate flowering phase, and it blooms throughout the growing season, which results in the strong competition of seeds for assimilates [3,5,9,12,24]. In the majority of ovules, egg apparatuses degenerate and legitimate pollen tubes do not reach the ovules [24].

The seed yield of common buckwheat is determined based on nectar production, due to its excellent composition for pollinators [14]. The reproductive success of other plant species also depends on the composition of the nectar. Buckwheat nectar contains sugars, mainly hexoses (glucose and fructose), sucrose, vitamins, and amino acids [17,18]. Although the primary pollinators of common buckwheat are honeybees, other insects, such as *Apis cerana*, *Bombus* spp., *Andrena* spp., *Osmia* spp., and *Diptera* species, also visit buckwheat flowers [9]. It is therefore obvious that the composition of the nectar is important for the frequency of visiting pollinators [8,14,19]. Plants evolutionarily adapted to pollinating insects offer them various compounds such as oils, nectar, resin, wax, pollen and fragrances; however, nectar is the most effective [18]. According to Brzosko et al. [18], the nectar of plants adapted to pollination by butterflies is characterized by high amino acid concentration, while that of plants pollinated by birds or flies is characterized by lower concentration. Nectar produced by different plant species has a different content of individual amino acids, from very high to their absence. Given that the bee population is declining as a result of chemicalization in agriculture, plants pollinated mainly by these insects will be forced to change the composition of the nectar, more suitable for other pollinators [17,19].

Currently, many stimulants are used in agriculture to improve crop quality and quantity. Stimulants can be of natural origin or produced by chemical synthesis. Most of the commercially available stimulants are mixtures of multiple active compounds that improve ion transport, nutrient uptake, and nutrient efficiency, stimulate photosynthesis, modulate phytohormones, and enhance tolerance to abiotic stress or boost crop quality and yield [22,25,26].

Plant hormones (auxins, gibberellins, cytokinins) stimulate the processes of pollination and seed setting [27]. Polyamines are classified as growth regulators and enhance stress tolerance, signaling, and gene expression [28,29]. Cysteine is an amino acid that contains an active –SH group and acts as an antioxidant, which is important considering that any environmental stress initiates oxidative stress, consisting in the formation of reactive oxygen species (ROS) [30]. Żur et al. [31] and Bianchi and Gibbs [32] used sodium chloride (NaCl) to break self-incompatibility in *Brassicaceae*. Commercial preparations stimulating plant growth and yielding are, for example, ASAHI SL (UPL) and TYTANIT^®^ (INTERMAG). ASAHI SL stimulates elongation growth and the development of generative organs, increases the accumulation of biomass and the intensity of photosynthesis, and protects plants against physiological stress [33]. This preparation is listed as harmless to the environment and is approved for use in organic farming [34]. TYTANIT^®^ is a liquid fertilizer containing titanium salts. It intensifies many biochemical processes, such as photosynthesis or transpiration, leading to faster above- and underground organ development [35]. 

As mentioned, in addition to the nutritional value for humans, an important ecosystem service associated with buckwheat cultivation is the maintenance of increased pollinator populations and biodiversity. The aim of this study was to investigate the effect of selected stimulants on the seed yield and nutritional value of buckwheat nectar. The latter issue has never been comprehensively studied despite the ever-increasing interest in the use of stimulants in agriculture and horticulture and the increasing attention paid to the protection and stimulation of biodiversity in agroecosystems. All three substances classified as biostimulants (ASAHI SL, TYTANIT^®^, NaCl) and plant hormones, which are not classified as biostimulants, were applied in this work. For this reason, the general term ‘stimulants’ was used [24]. 

The plants were treated exogenously with auxin NAA (1-naphthaleneacetic acid), cytokinin BAP (6-benzylaminopurine), cysteine, gibberellic acid (GA_3_), putrescine, NaCl (sodium chloride), ASAHI SL, and TYTANIT^®^. The stimulants were applied at either the beginning of flowering or three weeks later at the full flowering stage. Then, we analyzed the nectar composition of the open flowers and flower and seed production.

## 2. Results

### 2.1. Mass, Volume, and Sugar Composition of the Nectar

Statistical analyses showed that the stimulant type affected the nectar mass, and interaction between the common buckwheat line and the stimulant used was observed. The volume of the nectar depended solely on the line and term of application. The total content of sugars was determined by line, stimulant, and time of application. An interaction between the time of application and the accession, as well as between the time of application and the stimulant, was also found (Table 1).

The buckwheat nectar included such compounds as glycerol, inositol, glucose, fructose, sucrose, and fructans, as well as amino acids and polyamines (Figure 1). In the control plants, all parameters of the nectar composition (volume, mass, sugar content), except for glycerol at the beginning of flowering (term I), were higher for the PA15 than for the PA16 line (Figure 1A–I; Appendix A). At the full flowering stage (term II), the amounts of glycerol, glucose, fructose, and sucrose in the nectar of PA16 plants were higher than in PA15 plants (Figure 1E–I; Appendix A). In the PA15 line, the composition of the nectar collected at term I was higher than at term II. A characteristic feature of this line was a higher content of glycerol at the blooming stage than at the beginning of flowering (Figure 1E,F; Appendix A). In the nectar of the PA16 line, at term II only, fructose amount was greater than at term I, while glycerol amount was the same at both flowering stages (Figure 1I; Appendix A). Among fructans, kestose and nystose were detected in both lines, but kestose amount was much higher than that of nystose (Figure 1I).

The stimulants more often improved the nectar composition in PA15 plants than in the PA16 ones (Figure 1). The nectar volume in the PA15 line was greater after NaCl application at the term I of flowering, while in the PA16 line, it was greater after NAA application at term II (Figure 1A, Appendix A). The nectar mass was greater only in the case of PA15 plants treated with NaCl at term II (Figure 1C). Most stimulants increased glycerol amount in the nectar, mainly in PA15 plants at term I (Figure 1E,F). The most positive effect of the stimulants was observed on glucose content in both lines treated at the full blooming stage. Generally, no considerable increase in fructose content was observed; only GA_3_ improved the amount of this sugar in the PA15 line when applied at the full flowering stage. Also in this line, sucrose amount increased after ASAHI SL, cysteine, and TYTANIT^®^ application at term I and significantly under cysteine and GA_3_ impact at term II. Of all the stimulants, GA_3_ most effectively increased the content of all sugars in the PA15 line but only at the full blooming stage.

### 2.2. Amino Acid Content

The total content of amino acids was only affected by the term of the stimulant application (Table 1). In the control plants of both studied lines, the total amount of amino acids was significantly greater in the nectar produced at the beginning of the flowering stage than at the full blooming stage (Appendix A). In both lines, the most abundant amino acids in the total pool of amino acids were glutamine (51.25% and 54.72% at term I in PA15 and PA16 plants, respectively) and serine (about 8% in both lines), while the least abundant one was methionine (0.06% and 0.18% in the PA15 and PA16 lines, respectively) (Figure 1J, Appendix A). The nectar of both lines at the term II of flowering had the highest content of glutamine (31.5% and 29.95% in PA15 and PA16 plants, respectively). Again, in both lines, the least abundant amino acids at this flowering stagewere methionine and proline (0.12% and 0.33%, respectively, in PA15 plants and 0.14% and 0.26%, respectively, in PA16 plants). Cysteine was not detected in the nectar of either accession.

As in the case of sugars, the content of amino acids in the nectar of the PA15 line increased more under the influence of stimulants than in the PA16 line (Figure 1J; Appendix A). The most effective stimulants increasing the amounts of amino acids in the nectar of PA15 plants were NAA (it improved the amount of 14 amino acids), cysteine (11 amino acids), and BAP and putrescine (8 amino acids each) applied at the term I of flowering. At term II, NaCl increased the amount of eight amino acids (Appendix A). At the beginning of flowering, the amino acids most sensitive to stimulant treatment included isoleucine, leucine, arginine, aspartic acid, and lysine, while at term II, the most responsive amino acid was glutamine. In the PA16 line treated at term I, putrescine increased the amount of 7 amino acids, while at term II, NAA improved the amount of 10 amino acids (Appendix A). The amino acids most sensitive to stimulant treatment were glutamic acid at both flowering stages and aspartic acid at the beginning of blooming (Appendix A). Alanine content increased only after the application of BAP, NAA, GA_3_, cysteine, and TYTANIT^®^ in the second phase of blooming. 

### 2.3. Polyamine Content

The polyamines detected in the nectar of both lines included putrescine, cadaverine, spermidine, and spermine. Putrescine content was unaffected only by the type of the stimulant, while the amount of cadaverine did not depend on the common buckwheat line (Table 1). Contrary to that, the line and term of the stimulant application determined spermine content, while the line and the interaction between the line and the term of application affected spermidine amount.

Spermidine was the most abundant polyamine in the nectar of control PA15 and PA16 plants at both phases of flowering, and its levels amounted to 76.6 nM cm^−3^ and 87.5 nM cm^−3^, respectively (Figure 1K, Appendix A). The second most abundant polyamine was cadaverine (54.7 and 59.3 nM cm^−3^, respectively). In contrast with the other polyamines, only putrescine amount was significantly lower in the nectar collected at the full flowering stage. Generally, the content of polyamines did not change under the treatment with the studied stimulants. It is worth noting that the treatment with putrescine did not increase the amount of this polyamine in the nectar.

### 2.4. Flower Number and Seed Yield

Analysis of variance showed that flower productivity and seed number produced by one plant depended on the studied line of common buckwheat (Table 2). The type of the stimulant influenced seed yield, while flower productivity was affected by the term of its application. In this case, an interaction between the line and the time of application was observed.

Flower productivity of the PA15 line increased after the application of NAA and cysteine at the beginning of blooming, while most of the stimulants, except for BAP and ASAHI SL, evoked this effect when used at the full flowering stage (Table 3). In contrast, the stimulants decreased the flower number of the PA16 line at term I of flowering, but intensified flowering was observed for all the stimulants, except for cysteine and TYTANIT^®^, applied during the full flowering phase. The number of empty seeds in the PA15 line decreased under BAP, GA_3_, and NaCl treatment at term I and ASAHI SL and TYTANIT^®^ at term II. All the stimulants reduced the number of empty seeds in the PA16 line after application at the beginning of flowering, while at the second stage of blooming, this effect was only noted for GA_3_, cysteine, ASAHI SL, and TYTANIT^®^. An increase in the seed yield of the PA15 line was observed for all the stimulants at the beginning of flowering. The highest seed number was obtained after treatment with ASAHI SL and TYTANIT^®^. No stimulant increased the seed yield of PA15 when applied at the full blooming stage. A similar effect was noted for the PA16 line at both blooming phases. None of the stimulants changed the mass of a thousand seeds in the PA15 line at either of the blooming phases. This parameter increased in the PA16 line solely in the presence of ASAHI SL applied at the first term and after cysteine application at the second term. At this term, applied TYTANIT^®^ reduced the number of seeds per PA16 plant twice compared to that of the control, but the mass of thousand seeds (MTS) was the same. This result suggests that although TYTANIT^®^ decreased the number of ripe seeds, they were larger and better filled. A negative correlation (r = −0.489; *p* < 0.05) was found between the number of flowers and the number of ripe seeds. The number of ripe seeds of both accessions correlated positively with the total amount of sugars (r = 0.488; *p* < 0.05) and amino acids (r = 0.471; *p* < 0.05) in the nectar.

## 3. Discussion

The low yield of buckwheat is the main reason for the decreasing cultivation area of this valuable plant. Breaking the genetic restraints resulting from heterostyly and the defective development of embryo sacs with commonly used breeding methods seems unattainable. In our research, we tried to assess whether it is possible to increase the yield of common buckwheat by applying commonly used growth regulators and commercial preparations. The course of embryogenesis and the abortion of common buckwheat flowers and fruits, in addition to the genetic control, must be also influenced by plant hormones and growth regulators and in particular by the relationship between the concentrations of these compounds. 

Plants strongly interact with the biotic environment throughout their life cycle, including the production of nectar that attracts pollinating insects. The composition of the nectar is important for the frequency of visiting pollinators, and for the plant, it is an essential element of pollination success [36]. According to Cawoy et al. [14], the nectar mass and composition depend on the flower type and plant age, and the first flowers in anthesis produce more seeds than the flowers developed later on [37]. Moreover, Taylor and Obendorf [38] and Halbrecq et al. [39] reported that the production of nectar and seed sets depend on the position of the flower in a raceme due to access to assimilates, while according to Farooq et al. [40], nectar production also depends on the weather conditions such as light intensity, temperature, water availability, and buckwheat genotype. Our results proved that the abundance of buckwheat nectar is higher at the beginning of flowering than at the full flowering stage. This fact seems consistent with the results by Cawoy et al. [37], who stated that seeds are formed at the first phase of flowering, and at the later phase, the flower can be a source of nectar for insects, especially for bees, but does not affect seed setting. Also in our experiments, the seed yield obtained from all treatments was higher after applying the stimulators at the first phase of flowering.

Improved sugar composition of the nectar after the application of the stimulants was observed in both studied accessions, mainly following the treatment with NAA, ASAHI SL, cysteine, and GA_3_ only at the full flowering stage. Particularly, we detected increased amounts of glucose after the application of most stimulants. However, higher amounts of individual sugars did not affect the total pool of sugars. The stimulants also increased amino acid content in the nectar, especially in the PA15 line at the beginning of flowering. The amount of some amino acids increased also after stimulation at the full blooming stage. The positive effect of individual stimulants on amino acid composition depended on the flowering stage. NAA increased the amounts of various amino acids at the beginning of flowering, while NaCl increased them at the later flowering stage. These results indicate that a change in the nectar composition is specific for each stimulant and depends on the blooming phase. In contrast, Petanidou et al. [41], who studied 73 plant species from the Mediterranean area, found that although the composition of the nectar was important for the frequency of pollinators, it was not significantly influenced by the taxonomy of the visiting species or the time of flowering. In addition, these authors concluded that the total sugars, contrary to total amino acids, did not shape much the interest of insects. They detected 22 amino acids in the nectar, of which 15 were common in the flowers. They further observed that arginine and tryptophan acted rather as repellents, and phenylalanine content was crucial for attracting bees. In our experiment, the nectar of both studied accessions of buckwheat contained very small amounts of phenylalanine and tryptophan, while the level of arginine was also relatively small in the entire pool of amino acids. However, most applied stimulants significantly increased arginine amount in the nectar of PA15 flowers at the beginning of the flowering stage. If arginine exhibited any repellent properties, they had no effect on seed yield, which increased significantly in this genotype under the influence of the stimulants. It is worth noting that the seed yield positively correlated with the total pool of sugars and amino acids, which is consistent with the result obtained by Petanidou et al. [41].

The nectar may also contain other secondary metabolites, for example, non-protein amino acids or alkaloids [36]. Singaravelan et al. [42] reported that low concentrations of nicotine and caffeine elicited a significant feeding preference in free-flying honeybees. Biochemical analyses of buckwheat nectar in our study did not confirm the presence of the above compounds. However, apart from sugars and amino acids, such polyamines as cadaverine, spermine, spermidine, and putrescine were detected. In the nectar of both studied accessions, spermidine was the most abundant. Although the polyamine content was significantly affected by genotype and flowering date, the content of individual polyamines at both flowering stages was specific. 

Unusually large amounts of the polyamine putrescine in the nectar of the *Datura wrightii* flower were detected by Cheng [43]. This research, however, did not show the role of polyamine in the relationship between the pollinator and the plant. In turn, Aloisi et al. [44] suggest that polyamines in pear and pummelo nectar are involved in temperature-dependent pollen–pistil interactions and self-incompatibility response. These data may explain the high content of polyamines in the nectar of common buckwheat, which is characterized by strong self-incompatibility and sensitivity to high temperature increasing the degeneration of embryo sacs and embryos [12,18]. 

Based on our observations, we can suppose that the applied stimulants did not have any adverse effects on the pollinators. Used in small amounts, they were more insect attractants than repellents. Under an open foil tunnel, we found the presence of various pollinators, such as flies, bumblebees, bees, and ants. Unfortunately, we did not conduct research on the frequency of pollinating insects on plants treated with stimulants. Such research would certainly be interesting and may be the subject of further experiments. 

Most of the stimulants increased the number of flowers produced by plants; however, a greater number of flowers did not increase seed yield, which was also reported by other researchers [17,37,39,40]. The seed yield of the PA15 line was improved following the treatment with all stimulants at the first stage of blooming, especially after treatment with ASAHI SL and TYTANIT^®^. In the PA16 line, the positive effect of all the stimulants was shown by the reduction in the number of empty seeds at the beginning of flowering, while ASAHI SL and TYTANIT^®^ were also effective when applied at the full blooming stage. Unfortunately, in this line, the reduction in empty seeds did not increase ripe seed yield. It should be noted that the mass of a thousand seeds of the PA16 line increased significantly only after the application of ASAHI SL at the first flowering date. This is consistent with our previous results based on the embryological observation of developing embryos showing that ASAHI SL and TYTANIT^®^ significantly (up to 12-fold) decreased embryo abortion in two other common buckwheat lines [24].

In this study, it is difficult to indicate one versatile stimulant; however, most of the substances used showed a positive effect on nectar content and production and total yield. This could be a promising direction in improving the yield of this valuable crop.

## 4. Materials and Methods

### 4.1. Plant Material and Cultivation

This study was performed on two Polish accessions of common buckwheat: lines PA15 (currently registered as cv. ‘Korona’) and PA16. PA15 line is characterized by higher flower production and seed yielding than PA16 one. The seeds were obtained from the Plant Breeding and Production Station in Palikije, branch of Malopolska Plant Breeding (Poland). The plants grew in an open foil tunnel of the University of Agriculture in Kraków (Poland) located at the latitude 50°04′10″ N and the longitude 19°50′44″ E. The seeds were sown into pots (20 cm × 20 cm × 25 cm), 5 plants per pot, and filled with a commercial substrate mixed 1:1 (*v*:*v*) with perlite. For each accession, the seeds were sown into 108 pots (540 plants of each accession). The plants of both lines of the same treatment were grouped. The experiment was set up in mid-May in an open foil tunnel, allowing the control of the humidity of the substrate, as well as free access of pollinators. The plants were fertilized with Hoagland medium [45] once a week. The ripe seeds were harvested at the end of August.

### 4.2. Stimulant Application

The stimulants were used in aqueous solution concentrations applied in the previous experiment [18]: GA_3_ 100 mg dm^−3^, NAA 50 mg dm^−3^, BAP 50 mg dm^−3^, putrescine 290 mg dm^−3^, cysteine 100 mg dm^−3^ (all hormones were obtained from Sigma-Aldrich, Poznań, Poland), ASAHI SL (UPL) 1 cm^3^ dm^−3^, TYTANIT^®^ (INTERMAG) 0.3 mg dm^−3^ (both preparations were obtained from Agrosimex.pl, Poland), and commercial NaCl 10 mg dm^−3^. ASAHI SL involves following compounds: sodium para-nitrophenolate (0.3%), sodium ortho-nitrophenolate (0.2%), and sodium 5-nitroguaiacolate (0.1%). TYTANIT^®^ is a liquid fertilizer containing 0.8% of titanium salts. The control plants were sprayed with distilled water. Each plant was sprayed with 100 cm^3^ of the solution. 

The stimulants were applied on two dates: at the beginning of flowering (when most plants produced flower buds) or three weeks later at full bloom. In total, the experiment included 36 objects (2 lines × 9 treatments × 2 spray dates). Each treatment for each line involved 30 plants. Two weeks after the application of the stimulants, the flowers were collected for quantitative and qualitative nectar analysis. 

### 4.3. Analysis of Nectar Composition

Nectar composition was analyzed in open flowers collected two weeks after the stimulant application: at the beginning of flowering and at full bloom, using high-performance liquid chromatography (HPLC) according to Hura et al. [46]. The nectar was obtained from 50 open flowers per plant; 5 plants per genotype. The flowers were collected in the morning between 8:00 and 10:00. All analyses of the nectar composition were performed in three replicates for each combination involving three plants of each line/stimulant.

#### 4.3.1. Preparation of Buckwheat Flower Nectar Samples

The flowers were centrifuged in microtube Costar Spin-X 0.22 µm nylon membrane filters (Corning, NY, USA) at 22,000× *g* (45 min, 15 °C, Hettich Universal 32R, Hettich, Tuttlingen, Germany). The collected nectar was accurately weighted on a microanalytical scale (MYA 21.4Y, Radwag, Radom, Poland), and its density was estimated based on 15 µL aliquots (used for sugar analysis). After that, the samples were kept for further analyses at −80 °C.

#### 4.3.2. Soluble Carbohydrates

Sugars were analyzed according to the method by Hura et al. [46]. Exactly 15 µL of the nectar was accurately weighted into 2 cm^3^ Eppendorf tubes, diluted with acetonitrile/water 1:1 (*v*/*v*) to 1.8 cm^3^, and analyzed by HPLC for the content of soluble sugars. HPLC analyses were performed using an Agilent 1200 system (Agilent Technologies, Waldbronn, Germany) coupled to an ESA Coulochem II 5200A electrochemical HPLC detector (ESA, Chelmsford, MA, USA). Separation of soluble sugars (glucose, fructose, sucrose, and other fructans) was carried out using an RCX-10, 7 µm, 250 mm × 4.1 mm column (Hamilton, OH, USA), in a gradient mode of 75 mM NaOH and 500 mM sodium acetate in 75 mM NaOH. The concentrations of glycerol and sugars, such as glucose, fructose, sucrose, maltose, kestose, and nystose, were determined using the standards purchased from Sigma-Aldrich (Poznań, Poland). 

#### 4.3.3. Amino Acids

The content of amino acids was analyzed by means of a liquid chromatography technique employing an online-pre-column derivatization according to the method of Schuster [47]. A total of 20 µL of the nectar was diluted to 100 µL with 0.1 M HCl containing norvaline and sarcosine as internal standards (ca. 250 nM cm^−1^ each). Primary amino acids were derivatized with ortho-phthalaldehyde, whereas secondary amino acids were derivatized with 9-fluorenylmethyl chloroformate following the protocol by Woodward et al. [48]. An Agilent Infinity 1260 UHPLC system with a diode array detector (DAD) and FLD and Poroshel 120 HPH 3.0 mm × 100 mm 2.7 µm analytical column were used. The separation was performed under a linear gradient of 10 mM Na_2_HPO_4_ and 10 mM Na_2_B_4_O_7_ (pH 8.2) versus ACN/MeOH/H_2_O (45:45:10; *v*/*v*/*v*). Absorbance at 338 nm and 232 nm and fluorescence emission at 450 nm upon excitation at 230 nm were measured. Quantitation was conducted based on calibration curves for pure amino acid standards, considering the recovery of internal standards (Sigma-Aldrich, Poznań, Poland). 

#### 4.3.4. Polyamines

Polyamines were measured according to a modified method of Hura et al. [49]. A total of 10 µL of the nectar was diluted to 400 µL with 6% (*v*/*v*) trichloroacetic acid and mixed with 400 µL of dansyl chloride solution (5 mg cm^−1^ in acetone) and 400 μL of saturated sodium carbonate solution. The samples were incubated at room temperature overnight. Dansylated polyamines were triple-extracted with toluene and evaporated under nitrogen. The dried residues were dissolved in 50 µL of methanol and analyzed in Agilent 1200 system with a fluorescence detector (FLD) and Poroshell 120 EC-C18 3.0 mm × 50 mm 2.7 µm analytical column (Agilent Technologies) under a linear gradient of water (A) and methanol/acetonitrile (2:1) (B), both diluted with 1% (*v*/*v*) HCOOH. Detection was conducted at 350 nm excitation and 510 nm emission wavelengths. Quantitation was performed based on calibration curves constructed for dansylated polyamine standards (Sigma-Aldrich, Poznań, Poland). 

### 4.4. Flower and Seed Production

Flower number counting was performed at the end of the full flowering phase on 20 plants of each accession per treatment. At the end of the growing season, when most of the seeds were matured, the number of empty and ripe seeds, their mass, and mass of a thousand seeds (MTS) were analyzed for 20 plants of each genotype per treatment. The seeds were recognized as empty (containing none at all or abortive embryos) based on our previous research [20]. Embryo dying was a result of starvation, i.e., the lack of sufficient amount of assimilates delivered to the developing embryo.

### 4.5. Statistical Analysis

Analysis of variance (two-way ANOVA) was performed using the STATISTICA 13 package (Statsoft, Tulsa, OK, USA). The significance of differences between means was analyzed according to Duncan’s multiple-range test at *p* < 0.05. The values represent means ± SE (standard error). Pearson’s correlation coefficients were assumed significant at *p* < 0.05.

## 5. Conclusions

The applied stimulants have specific effects on the composition of buckwheat nectar, the process of flowering, and seed production. We also showed that each accession responded individually to the substances used. Some of the studied parameters increased after the stimulant treatment at the initial phase of flowering, and some increased only at a later stage. In most cases, sugar content in the nectar and flower production can be improved by the application of stimulants at the full flowering stage, while seed yielding enhances after the treatment at the beginning of bloom. The higher total amount of sugars and amino acids in the nectar of common buckwheat significantly determined seed yield. All the stimulants used reduced the number of empty seeds in both accessions. Although we were unable to select one universal stimulator that would improve the composition of the nectar and seed yield, seed production in the PA15 line increased significantly under the influence of all stimulants used at the beginning of flowering, and the most effective were ASAHI SL and TYTANIT^®^. The results of the study indicate that the use of stimulants in buckwheat cultivation can influence both its nectar value for pollinating insects and yield. Unfortunately, the observed interactions between the type of stimulant and the line indicate the need to select the optimal cultivated variety. Analyzing the results obtained in our experiments, we can recommend both ASAHI SL and TYTANIT^®^ preparations as the most promising in improving the seed yield of common buckwheat. Further research on the impact of stimulants on the physiological and metabolic features of crop plants should also include observations on the frequency of fly-bys of pollinating insects.

## Figures and Tables

**Figure 1 ijms-24-12852-f001:**
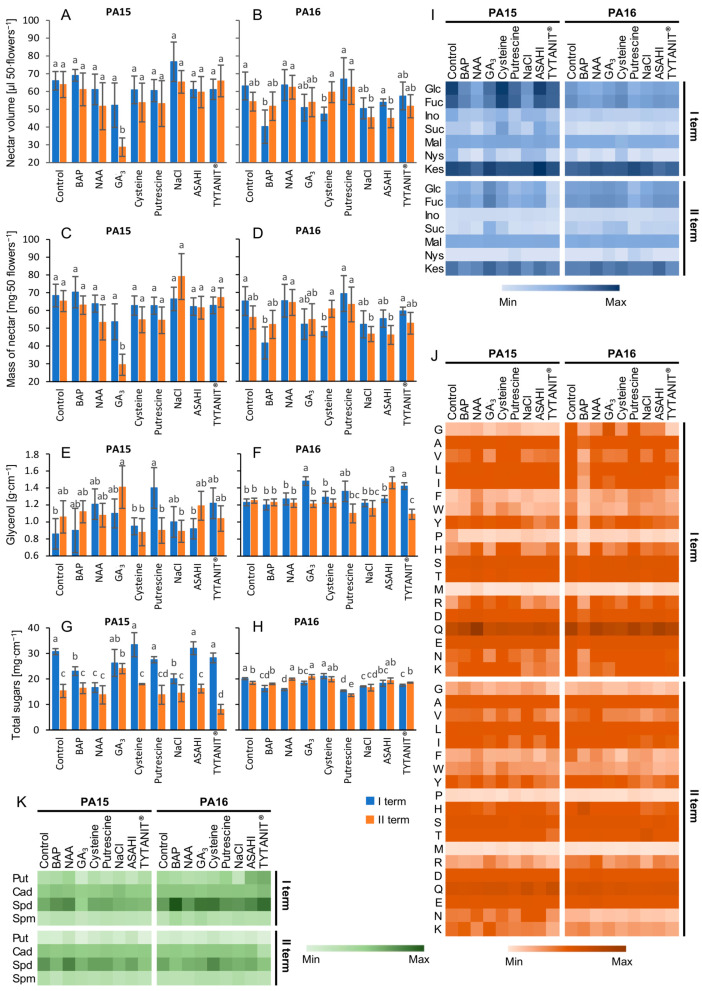
Effects of the stimulants applied at the beginning of flowering (term I) or at full flowering stage (term II) on the volume, mass, and composition of the nectar in the flowers of the PA15 and PA16 lines of common buckwheat. (**A**,**B**) Nectar volume. (**C**,**D**) Nectar mass. (**E**,**F**) Glycerol content. (**G**,**H**) Total sugar content. (**I**) Heat map of sugar composition. (**J**) Heat map of amino acid composition. (**K**) Heat map of polyamine composition. The values shown in the charts are means ± SE (*n* = 3). Different letters indicate significant differences (*p* < 0.05, Duncan’s multiple-range test). Amino acid symbols are explained in Abbreviations; Glc—glucose; Fuc—fructose; Ino—inositol; Mal—maltose; Nys—nystose; Kes—kestose; Put—putrescine; Cad—cadaverine; Spd—spermidine; Spm—spermine.

**Table 1 ijms-24-12852-t001:** Analysis of variance (Test F) of the nectar composition: mass, volume, total content of sugars and amino acids, content of putrescine (Put), cadaverine (Cad), spermidine (Spd), and spermine (Spm) in lines PA15 and PA16 of common buckwheat plants treated with various stimulants applied at the beginning of flowering or in the full flowering phase; *—*p* < 0.05; **—*p* < 0.01; ***—*p* < 0.001; ns—not significant.

Effects	Mass	Volume	Sugars	Amino Acids	Put	Cad	Spd	Spm
Line (L)	ns	***	***	ns	**	ns	**	***
Stimulant (S)	*	ns	***	ns	ns	***	ns	ns
Term of application (T)	ns	ns	***	***	***	***	**	ns
L × S	*	ns	ns	ns	**	***	ns	ns
L × T	ns	*	**	ns	***	***	***	**
S × T	ns	ns	***	ns	***	***	ns	ns
L × S × T	ns	ns	ns	ns	***	***	ns	ns

**Table 2 ijms-24-12852-t002:** Analysis of variance (Test F) on the flowering and seed productivity of PA15 and PA16 common buckwheat lines treated with various stimulants applied at the beginning of flowering or at the full flowering phase; *—*p* < 0.05;; ***—*p* < 0.001; ns—not significant.

Effects	Flower Productivity	Seed Yield
Line (L)	***	***
Stimulant (S)	ns	***
Term of application (T)	***	ns
L × S	*	ns
L × T	ns	ns
S × T	ns	*
L × S × T	*	ns

**Table 3 ijms-24-12852-t003:** Flower, empty, and ripe seed number per plant and the mass of a thousand seeds under the stimulant application at the beginning of flowering (term I) or at the full flowering (term II) of the PA15 and PA16 lines of common buckwheat. Means (*n* = 20) marked with the same letter do not differ according to multiple-range Duncan’s test (*p* < 0.05). MTS—mass of a thousand seeds.

Stimulant	PA15				PA16			
	No. of Flowers	No. of Empty Seeds	No. of Ripe Seeds	MTS	No. of Flowers	No. of Empty Seeds	No. of Ripe Seeds	MTS
Term I								
Control	616.1 c	22.6 a	117.9 c	30.86 a	699.3 a	21.6 a	145.7 a	28.68 b
BAP	632.7 c	16.0 b	133.4 b	33.24 a	444.4 c	14.7 b	134.1 b	28.00 b
NAA	715.3 b	18.1 ab	130.5 b	30.30 a	413.7 c	14.9 b	133.6 b	29.97 b
GA_3_	624.5 c	13.5 b	130.0 b	30.50 a	539.7 b	9.8 c	137.5 b	26.37 b
Cysteine	876.0 a	28.0 a	160.0 a	28.33 a	490.2 bc	11.9 b	139.0 b	28.27 b
NaCl	630.0 c	15.5 b	132.0 b	28.62 a	461.0 c	7.0 c	149.9 a	29.12 b
ASAHI SL	653.7 c	25.7 a	168.9 a	30.83 a	398.4 cd	13.3 b	98.8 c	38.22 a
TYTANIT^®^	657.7 c	25.5 a	162.8 a	31.84 a	500.4 b	11.0 b	110.9 c	27.13 b
Putrescine	466.8 d	23.4 a	138.6 b	32.71 a	451.0 c	16.0 b	148.1 a	26.53 b
Term II								
Control	616.1 e	22.6 b	117.9 a	30.86 a	699.3 c	21.6 a	145.7 a	28.68 b
BAP	610.6 e	23.9 bc	112.1 a	29.94 a	880.6 a	17.1 a	101.3 c	27.55 bc
NAA	932.4 c	27.2 b	107.2 b	31.54 a	900.2 a	19.7 a	130.0 b	26.44 c
GA_3_	1072.8 b	26.1 b	95.6 c	31.83 a	807.8 b	13.7 b	116.3 c	27.70 bc
Cysteine	935.8 c	25.9 b	104.4 b	29.35 a	564.4 d	14.3 b	84.2 d	31.00 a
NaCl	933.4 c	28.1 b	100.7 bc	30.41 a	833.8 b	18.0 a	108.9 c	27.35 bc
ASAHI SL	512.8 f	18.2 c	85.3 d	34.08 a	917.2 a	14.8 b	103.2 c	28.72 b
TYTANIT^®^	1268.8 a	16.2 c	90.6 cd	28.17 a	550.6 d	14.2 b	67.5 e	29.17 ab
Putrescine	910.6 c	36.8 a	99.7 c	32.63 a	822.7 b	18.7 a	107.4 c	27.72 bc

## Data Availability

Not applicable.

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
