# Peer review of "The Effect of Stimulants on Nectar Composition, Flowering, and Seed Yield of Common Buckwheat (Fagopyrum esculentum Moench)"

_ijms, 2023, doi:10.3390/ijms241612852_

Round 1

Reviewer 1 Report

I want to express my pleasure for being chosen to review the manuscript titled: “The Effect of Stimulants on Nectar Composition, 2 Flowering, and Seed Yield of Common Buckwheat 3 (Fagopyrum esculentum Moench)
From my review of the manuscript, it became clear to me that the research point was dealt with by the authors very accurately, but there are some comments that must be remedied before publication.
In my opinion, the authors have collected a unique dataset using cutting edge methodology. But I would like to offer few suggestions so that authors should improve it before publication.
Ø Abstract section: In the abstract, the research gap should be improved to strengthen the motivation of the work and considering the important results.

-The Abstract lacks any of the improvements that led to an improvement in production
Ø The novelty of the study needs to be highlighted compared to other similar studies.

- Add all materials used in Materials section.

Results section . Histograms seem unclear and incomprehensible. They should be presented clearly

Introduction part:
Ø Must contains the whole background regarding the targeted problem and how to solve that problem with comparison with literature review; please check and revised accordingly.
Ø Authors should choose one style for writing references in the manuscript.

The English language needs to be significantly improved, in wording, grammar and sentence structure.

I suggest the authors to go through the manuscript one more time to minimize some errors, typos etc.

Conclusion section: it is good to include some recommendations.

Reviewer 2 Report

Thank you for this exciting study.

The chemical analyses are well conducted and reasonably complete. The results are well-presented and significant. The paper is easy to read. However, I do not understand why you focus so much on the nectar production of this crop when you do not assess the impact of the composition of the nectar on the entomophilous pollination service rendered to this crop.

I think the narration could be reworked a bit:

- you seek to improve the yield by fertilizing to reduce the number of aborted seeds

- you show in this paper that this fertilization also has a marked impact on the composition of the nectar

- Is there a pollination deficit that could be reduce improving the insect pollination? Would these modifications of the nectar have an impact on the insects, which could explain the differences in yield?

- In a future study, it would be interesting to study the impact of these nectar modifications on entomophilous pollination services.

You never evaluate or provide a figure to estimate the dependence of this crop on pollinating insects, and you never check whether or not there is a pollination deficit, which could come from a deficit of insect attractiveness. Moreover, you have not studied the diversity of insects that visit buckwheat nor the impact of the variation in the composition of the nectar on these visits. L458-459, you explain, based on your previous research, that the yield deficit seems to be linked to the difficulty of the plants in nourishing the embryo, which leads to empty seeds. Pollination has therefore been carried out, and the challenge is no longer to attract more insects but to provide fertilization which allows the plants to have enough nutrients to fill the seeds! It's really interesting to study the impact of fertilization on nectar quality. Still, I'm a bit concerned about the link you imply between nectar composition and seed yield, which would imply more entomophilous pollination.

You have never tested the impact of nectar enrichment on insect attractiveness and pollination quality. I think in this paper, this aspect should only be addressed in the discussion.

I have the impression that your study seeks, above all, to show that by reducing the nutritional deficiencies of the plant, we improve the yield by reducing the number of aborted seeds, increasing the number of flowers and improving the MTS; and that one of the collateral impacts is a change in the composition of the nectar. In discussion, we could then wonder about the impact of these modifications of the nectar on the attractiveness of insects and the quality of the entomophilous pollination service, which results therefrom, which would make another great study to conduct!

Please find some more specific comments below.

L. 26 It is unclear why you want to focus on increasing the nectar production; in this paper, you never observe the impact of nectar production on insects attractively or pollination services. I guess that the real aim is only to increase the seed yield. But fertilising the plants to increase seed yield also increases nectar secretion and composition. The next step (for another paper) will be to try to see if the yield increase is a consequence of the improved nectar secretion or not. 

L. 31-32 Can you precise what kind of correlation? (Positive, negative?)

L. 81-82 You present honeybees as the primary pollinator of common buckwheat. Can you provide references and maybe also quantify the insect pollination dependency of the crop? If the insects play a main role in pollination and if there is an apparent pollination deficit for this crop, that would help to understand the interest of your study. But, beyond that, is the lack of pollination linked to a lack of attractiveness of the crop or only to a lack of available insects? Would the improved nectar secretion have an impact on a commercial field?

L. 95 "compound" or products?

L. 96 ASAHI SL and TYTANIT. Are they commercial names? Are they patented, so do they need a (R)? What are the firms that produce them? What are their compositions, and what are the recommendations for use?  Even in the Material and Method section, you provide no presentation of these products; we need more information.

L. 300-301 positive or negative correlation?

L. 328-330 "the applied stimulators did not have any adverse effects on the pollinators." Please provide references or figures.  

L. 332-334 "The studied stimulants influenced rather flower production than nectar composition in both accessions, and they showed their effects rather at the full blooming stage." Isn't it contradictory to your previous results?

L.369-373 Can you explain why you choose these concentrations?
